# Neurotrophin Signaling in Medulloblastoma

**DOI:** 10.3390/cancers12092542

**Published:** 2020-09-07

**Authors:** Amanda Thomaz, Mariane Jaeger, Algemir L. Brunetto, André T. Brunetto, Lauro Gregianin, Caroline Brunetto de Farias, Vijay Ramaswamy, Carolina Nör, Michael D. Taylor, Rafael Roesler

**Affiliations:** 1Cancer and Neurobiology Laboratory, Experimental Research Center, Clinical Hospital (CPE-HCPA), Federal University of Rio Grande do Sul, Porto Alegre 90035-003, RS, Brazil; a.thomaz@lancaster.ac.uk (A.T.); labpesquisa1@ici.ong (M.J.); institucional@ici.ong (A.L.B.); andrebrunetto@ici.ong (A.T.B.); lgregianin@hcpa.edu.br (L.G.); labpesquisa@ici.ong (C.B.d.F.); 2Department of Pharmacology, Institute for Basic Health Sciences, Federal University of Rio Grande do Sul, Porto Alegre 90050-170, RS, Brazil; 3Children’s Cancer Institute, Porto Alegre 90620-110, RS, Brazil; 4Department of Pediatrics, School of Medicine, Federal University of Rio Grande do Sul, Porto Alegre 90035-003, RS, Brazil; 5Pediatric Oncology Service, Clinical Hospital, Federal University of Rio Grande do Sul, Porto Alegre 90035-003, RS, Brazil; 6The Arthur and Sonia Labatt Brain Tumour Research Centre, The Hospital for Sick Children, Toronto, ON 17-9702, Canada; vijay.ramaswamy@sickkids.ca (V.R.); carolina.nor@sickkids.ca (C.N.); mdtaylor@sickkids.ca (M.D.T.); 7Division of Haematology/Oncology, The Hospital for Sick Children, Toronto, ON M5G 1X8, Canada; 8Developmental and Stem Cell Biology Program, The Hospital for Sick Children, Toronto, ON M5G 1X8, Canada; 9Department of Laboratory Medicine and Pathobiology, University of Toronto, Toronto, ON M5S 1A1, Canada; 10Division of Neurosurgery, The Hospital for Sick Children, Toronto, ON M5G 1X8, Canada

**Keywords:** nerve growth factor, brain-derived neurotrophic factor, tropomyosin receptor kinase, neurotrophin, medulloblastoma, brain tumor

## Abstract

**Simple Summary:**

Neurotrophins are a family of proteins known for regulating nervous system development and neuronal survival and plasticity. These proteins act by activating specific receptor proteins on the cell surface. More recently, neurotrophins and their receptors emerged as mechanisms contributing to cancer progression. Cancer is the most common cause of disease-related death in children. Here, we review the evidence indicating a role for neurotrophin-mediated cell signaling in medulloblastoma, the most common type of malignant brain cancer of the childhood. In addition, by analyzing gene transcript profiles in datasets of tumors from patients with medulloblastoma, we revealed novel findings supporting neurotrophin receptors as potential molecular prognostic markers of patient survival.

**Abstract:**

Neurotrophins are a family of secreted proteins that act by binding to tropomyosin receptor kinase (Trk) or p75NTR receptors to regulate nervous system development and plasticity. Increasing evidence indicates that neurotrophins and their receptors in cancer cells play a role in tumor growth and resistance to treatment. In this review, we summarize evidence indicating that neurotrophin signaling influences medulloblastoma (MB), the most common type of malignant brain cancer afflicting children. We discuss the potential of neurotrophin receptors as new therapeutic targets for the treatment of MB. Overall, activation of TrkA and TrkC types of receptors seem to promote cell death, whereas TrkB might stimulate MB growth, and TrkB inhibition displays antitumor effects. Importantly, we show analyses of the gene expression profile of neurotrophins and their receptors in MB primary tumors, which indicate, among other findings, that higher levels of *NTRK1* or *NTRK2* are associated with reduced overall survival (OS) of patients with SHH MB tumors.

## 1. Introduction

Driver oncogenes in many types of cancer encode growth factor receptors belonging to the receptor tyrosine kinase (RTK) superfamily, and most molecularly targeted therapies successfully incorporated into clinical practice to date act by targeting RTKs [1,2]. Neurotrophins, protein growth factors that activate RTKs of the tropomyosin receptor kinase (Trk) family, are known to be critically involved in regulating neuronal development and have also been increasingly implicated in tumor progression and resistance to treatment in several types of cancer, including those of possible neural origin [3,4,5]. In children, brain cancers are the most common solid tumors and the leading cause of cancer-related mortality. The most common type of malignant childhood brain tumor is medulloblastoma (MB). Conventional multimodal treatment with chemotherapy, radiotherapy, and surgery has improved cure rates over the last decades, but unfortunately, about one-third of patients will relapse. Survivors may often experience long-term neurological, cognitive, and endocrinological deficits secondary to curative intent [6,7]. Smarter treatment modalities are needed to improve outcomes and reduce morbidity. Here, we review the emerging evidence indicating that neurotrophin signaling is involved in MB pathogenesis, discuss some of the potential biological, prognostic and clinical implications and propose the neurotrophin/Trk pathway as a promising target in MB treatment.

## 2. MB Biology

### 2.1. Molecular Subgroups of MB

Some of the most important advances in our understanding of MB are related to its classification into four consensus molecular subgroups with distinct genomic, epigenetic, and clinical features: WNT, SHH, Group 3, and Group 4 [7,8,9]. This classification has quickly become critically important for guiding patient risk stratification, treatment, and selection in clinical trials [9,10]. The WNT and SHH subgroups are defined by mutations leading to aberrant activation of the Wingless and Sonic hedgehog pathways, respectively, whereas Group 3 MB has been associated with amplification of genes involved in the Notch and transforming growth factor β (TGFβ) pathways, and Group 4 with an increased representation of genes involved in chromatin modification [11,12]. Patients who exhibit tumors of the more aggressive Group 3 and Group 4 subgroups have a particularly poor prognosis, with patients with Group 3 MB showing a 5-year survival of around 50% and a high rate of metastasis at diagnosis [9,10,13]. More recently, significant intra- and intertumoral heterogeneity has been reported within specific subgroups, and at least 12 unique subtypes within subgroups have been identified [7,14,15,16,17,18].

### 2.2. MB Origins

Identifying the cellular origin of MB is crucial to understand how normal cells transform into cancer cells. Several childhood tumor types are believed to emerge from errors in development, either directly from embryonal cells or from more mature prenatal cell types that acquire embryonal properties, including aberrant self-renewal capacity [19,20]. In MB, embryonic neural stem cells (NSCs) and different types of neural precursors have been proposed as candidate cells of origin [21,22].

Different molecular subtypes of MB mirror fetal transcription programs from distinct cerebellar cell lineages that may appear transitionally during development [23]. Cell types that descend from cerebellar stem cells, including typical and Nestin-expressing progenitors (NEPs) committed to the granule neuron lineage, can give rise to SHH-subtype MB upon genetic ablation of *Ptch 1*, which encodes the Sonic hedgehog receptor protein patched homolog 1 [24,25]. SHH MB can also arise from a rare and transient Sox2^+^ GNPC population [26]. Different embryonic cerebellar progenitor cells likely originate Group 3 MB [27,28], and deep cerebellar nuclei (DCNs) or their precursors are among the candidate cells of origin of Group 4 MB [29]. In contrast, WNT-subtype MB can originate outside the cerebellum, from neuron precursors of the dorsal brainstem [30]. Recent findings indicate that SHH MB transcriptionally resembles the granule cell hierarchy, whereas Group 3 MB mirrors Nestin+ stem cells, and Group 4 MB resembles unipolar brush cells. In addition, bulk tumors can contain a mixed population of transcriptionally distinct cells [23].

## 3. Neurotrophin Regulation of Nervous System Development and Function

Multiple signaling pathways involved in normal neural development and plasticity are hijacked and amplified by cancer to promote tumor growth. Neurotrophins and their receptors play a critical role in regulating nervous system development and neuronal survival and plasticity. Trk neurotrophin receptors, TrkA (encoded by the *NTRK1* gene), TrkB (encoded by *NTRK2*), and TrkC (encoded by *NTRK3*) are activated primarily by their endogenous ligands, nerve growth factor (NGF), brain-derived neurotrophic factor (BDNF), and neurotrophin 3 (NT-3), respectively. NT-3 also activates TrkA, and TrkB can also be activated by NT-3 and neurotrophin NT-4/5 (NT-4/5). Trk activation by neurotrophins induces receptor homodimerization and transphosphorylation of critical tyrosine residues, leading to intracellular signal transduction mediated by activation of multiple protein kinase pathways, including phosphoinositide 3-kinase (PI3K), mitogen-activated protein kinase (MAPK), and phospholipase C-gamma (PLCγ)/protein kinase C (PKC), ultimately resulting in cAMP response element-binding protein (CREB) phosphorylation and changes in gene expression. Other intracellular mechanisms mediating the actions of neurotrophins include increased synaptic insertion of α-amino-3-hydroxy-5-methyl-4-isoxazolepropionic acid (AMPA) glutamate receptors, through a mechanism dependent on PKC and Ca2+/calmodulin kinase II (CaMKII), as well as modulation of voltage-gated ion channel function. Neurotrophins and proneurotrophins also act by binding the p75 neurotrophin receptor (p75NTR, also known as CD271), a member of the tumor necrosis factor receptor superfamily, to promote either the activation of nuclear factor-kappa B (NF-kB) signaling pathway, and induce prosurvival signals, or activate the c-Jun N-terminal kinases (JNK) signaling pathway to generate cell death [31,32,33,34]. Mechanisms of neurotrophin-independent Trk activation, for instance through transactivation dependent of G protein-couple receptors (GPCRs), have also been described [35].

Trks and p75NTR are expressed in embryonic NSCs, and NGF, BDNF, or NT-3 promote NSC survival [36,37]. In contrast, expression of TrkA or TrkC in the absence of neurotrophins triggers cell death in embryonic stem cells [38]. Thus, changes in NGF availability may crucially regulate the survival or naturally occurring death of neurons during development, particularly in the peripheral nervous system [39]. TrkB is mostly expressed in central nervous system (CNS) neurons, does not trigger cell death, and its stimulation by BDNF can stimulate the survival or differentiation of neurons derived from NSCs [36,38,39,40]. Neurotrophin signaling is also crucially involved in mediating activity-dependent refinement of neural circuits during development [41]. After neuronal differentiation and throughout adulthood, neurotrophins promote neurite outgrowth, axon formation, synapse maturation, long-term plasticity, memory, neuronal survival, and resistance to stress [31,32,33]. A summary of selected signaling pathways mediating the actions of neurotrophins is shown in Figure 1.

## 4. Neurotrophin Signaling in Cancer

Trk was originally identified as an oncogene in a colon cancer sample. Specifically, the altered gene consisted of an *NTRK1* gene fusion containing sequences from non-muscle tropomyosin (*TPM3*) [43]. The recurrence of this *TPM3-NTRK1* gene fusion as an oncogene in colon cancer has been more recently confirmed, along with evidence that it is associated with sensitivity to TrkA inhibition [44]. *NTRK1* fusions are now known to occur in many other solid tumor types, including lung adenocarcinoma, papillary thyroid carcinoma, secretory breast carcinoma, and glioblastoma (GBM) [45,46]. In addition, evidence indicating that NGF/TrkA, BDNF/TrkB, TrkC, or p75NTR play a role in cancer has rapidly accumulated over the past few years, with most studies showing that neurotrophins and their receptors are expressed in cancer cells and influence experimental tumor growth, cellular survival, proliferation, migration, invasion, neovascularization, metastasis, and treatment resistance in many peripheral solid tumor types including colorectal, breast, small cell and non-small cell lung, cervical, bladder, gallbladder, laryngeal, renal, head and neck, and oral squamous cell cancers [3,5,47,48,49,50,51,52,53,54,55,56,57,58,59,60,61].

Neurotrophins and Trk receptors also play a role in brain tumor types other than MB. Expression of NGF and BDNF has been found in samples of human astrocytoma [62]. Human malignant glioma samples and cancer stem cells (CSCs) isolated from human gliomas express NGF, BDNF, NT3, TrkB, and TrkC. Neurotrophin activation of TrkB and TrkC enhanced CSC viability through a mechanism dependent on the extracellular-regulated kinase (ERK) and Akt pathways. Conversely, knockdown or pharmacological inhibition of TrkB and TrkC decreased glioma CSC growth [63]. TrkA and TrkB can be activated in GBM cells, and combined inhibition of Trk and c-Met reduces the resistance against CDK4/6 inhibition in experimental GBM [64]. Selective TrkB inhibition effectively and dose-dependently impairs the viability of human GBM cells in vitro [65]. A systematic screening of a library of human tyrosine kinases for their oncogenic potential in glioma and found compelling evidence indicating that TrkB plays a role in tumor formation [66]. Furthermore, TrkB-containing exosomes in GBM cells can promote the transference of tumor aggressiveness among cells [67].

In pediatric solid tumors, the role of neurotrophin signaling has been mostly investigated in neuroblastoma (NB), a cancer type derived from embryonal neural crest cells that later give rise to the sympathetic nervous system and accounts for around 15% of pediatric cancer deaths [68]. NB tumors expressing high levels of TrkA show a favorable prognosis, whereas BDNF and TrkB expression is associated with worst outcomes [69,70]. TrkB stimulation by BDNF protects TrkB-expressing human NB cell lines against cytotoxic chemotherapeutics, and the protective effect of BDNF is prevented by inhibition of TrkB or PI3K [71,72]. BDNF protects NB cells from paclitaxel by downregulating the proapoptotic protein Bim through a mechanism dependent on MAPK [73]. BDNF has also been shown to stimulate, and NGF to inhibit, NB cell invasion [74], and BDNF activation of TrkB promotes metastasis in experimental NB through the PI3K and MAPK pathways [75]. In contrast, TrkA activation by NGF decreases N-myc expression through MAPK signaling, resulting in a reduction in the number of NB cells, and promotes NB cell differentiation [76]. p75NTR can induce apoptosis in NB cells and TrkA inhibits this effect [77,78], and p75NTR expression enhances the cytotoxic effect of the redox-active chemotherapeutic drug fenretinide in NB [79]. When co-expressed with TrkA and TrkB, p75NTR enhances Trk receptor sensitivity to low levels of ligand [80].

In Ewing sarcoma (ES), another type of pediatric solid tumor with possible origin in embryonal neural crest cells [81], treating human ES cells with TrkA or TrkB selective inhibitors reduced cell proliferation, and the effects were optimized when the two inhibitors were combined. Moreover, the pan-Trk inhibitor K252a induced changes in morphology, reduced levels of β-III tubulin, and decreased mRNA expression of NGF, BDNF, TrkA, and TrkB in ES cells, in addition to potentiating the effects of cytotoxic chemotherapy even in chemoresistant ES cells [82]. The possibility that Trk receptors are involved in carcinogenesis in tumors derived from embryonal neural crest cells is supported by evidence that constitutive activation of TrkB is sufficient to promote malignant transformation, accompanied by increased expression of MYCN and other cancer-associated genes and reduced expression of tumor suppressor genes, in neural crest cells. Importantly, neural crest cells with constitutively active TrkB form rapidly growing and invasive tumors when injected into NOD SCID mice [83].

In brain tumor types that afflict children (other than MB), *NTRK* mutations have been reported in pediatric low-grade and diffuse high-grade gliomas [63,84,85,86]. Activating fusions of *NTRK1*, *NTRK2*, or *NTRK3* occur in approximately 40% of pediatric high-grade gliomas and *NTRK2* fusions in about 3% of pediatric pilocytic astrocytomas [63,87,88]. To date, such genetic alterations in *NTRK* genes have not been reported in MB. TrkA and p75NTR are expressed in ependymoma [89]. NGF expression was reported to be increased, whereas BDNF expression was reduced, both in tumor samples and cerebrospinal fluid (CSF), in children with low-grade astrocytomas and ependymomas [90]. Although that study did not find changes in plasma levels of neurotrophins, recent findings in children with acute leukemia suggests that BDNF should be further investigated as a potential biomarker in pediatric cancers [91].

## 5. Neurotrophins and Their Receptors in MB

### 5.1. NGF and TrkA

Early studies in the 1990s, aimed at investigating the protein expression of neurotrophins and neurotrophin receptors in MB, were done at a time when MB was classified as type of primitive neuroectodermal tumor (PNET), thus MB samples were analyzed together with other PNETs. TrkA was found in 5 out of 20 and NGF in 6 out of 20 MB specimens, although NGF and TrkA were not expressed within the same tumors [92]. Washiyama et al. [93] found TrkA immunoreactivity in cells from 8 of 29 PNET samples, 27 of which were posterior fossa pediatric MBs. An immunohistochemical study focusing on neuronal differentiation in the nodules of nodular/desmoplastic MBs found that TrkA and NGF were expressed in 13 of 14 tumor samples, and were mostly localized within nodules, which is consistent with a role for Trks in apoptosis and neuronal differentiation in MB [94]. In a more recent study, TrkA was identified in 14 of 21 cases of pediatric patients with newly diagnosed MB, and TrkA expression was correlated to the apoptotic index [95].

NGF and TrkA were also identified in MB cell lines [96]. As mentioned above, although neurotrophin signaling is generally associated with cell survival, both TrkA and TrkC promote neuronal death in the developing nervous system [38]. Experimental studies using cultured cells have consistently supported the hypothesis that TrkA activation by NGF stimulates cell death and hinders growth in MB [47,97,98], and TrkA expression in MB is associated with neuronal differentiation, low proliferation, and apoptosis [99]. Treating MED-H MB cells with NGF resulted in growth inhibition and increased differentiation [95]. D283-MED and DAOY MB cells engineered to overexpress TrkA undergo apoptosis when treated with NGF, an effect that is blocked by anti-NGF antibodies or the pan-Trk inhibitor K252a [100]. The TrkA-mediated apoptosis is blocked by mutations in the ATP binding site or tyrosines 490 and 785. In addition, expression of a dominant negative Ras inhibitor prevents NGF-induced ERK activation and apoptosis, whereas ERK inhibition alone does not affect apoptosis [101]. NGF-induced apoptosis is associated with a reduced expression of the DNA-damage-inducible gene gadd45, possibly through BRCA1 and independently of c-Jun NH2-terminal kinase (JNK) or p38 MAPK [102]. NGF was found to impair proliferation and increase TrkA expression, but also to reduce the cytotoxic effect of cisplatin in MB cells [103]. Ectopic expression of the transcription factor Zhangfei in ONS-76 MB cells resulted in increased expression of TrkA and apoptosis markers [104]. Zhangfei enhances expression of Brn3a, an inducer of TrkA expression, promoting autocrine NGF stimulation of TrkA that leads to MAPK-dependent neuronal differentiation and cell death in ONS-76 MB cells [105]. In addition to promoting apoptosis and autophagy, NGF activation of TrkA can lead to cell death through casein kinase 1 (CK1)-mediated stimulation of macropinocytosis [106], which involves inhibition of RhoB and FRS2-scaffolded Src and H-Ras activation of RhoA [107]. Moreover, TrkA-induced cell death in MB cells depends on the cerebral cavernous malformation 2 (CCM2) protein and can be blocked by inhibition of the germinal center kinase class III kinase and CCM2 interactor STK25, but not STK24 [97,98,108].

### 5.2. BDNF and TrkB

An early immunohistochemical study detected the presence of BDNF and TrkB in 8 out of 20 MB samples, with co-expression of both molecules in 6 of the cases [92]. Another study found BDNF- and TrkB-positive tumor cells in 6 out of 27 and 18 out of 29 samples, respectively [93]. MB cell lines express both BDNF and TrkB [109,110], and TrkB blockade by the selective inhibitor ANA-12 can induce a pronounced inhibition of survival and viability, as well as cell cycle arrest, in cell lines (D283-MED and UW-228) associated with different MB molecular subgroups. In addition, we have recently shown that TrkB inhibition slows the growth of D283-MED MB tumors xenografted into nude mice *in vivo*, increased apoptosis, reduced ERK activity, increased expression of signal transducer and activator of transcription 3 (STAT3), and resulted in differential modulation of p21 expression [111] (Figure 2). However, TrkB activation by BDNF may also reduce cell viability under certain experimental conditions, either when given to MB cells alone [109] or combined with a histone deacetylase inhibitor [112]. It is worth noting that differences in results obtained with different cell lines may be related to distinct origins and biological features. For example, D283-MED cells, which produce tumors in mice that respond to TrkB inhibition [111], are representative of Group 3/4 MB and derive from a metastatic site [113]. Studies have begun to uncover molecular differences between MB metastases and primary tumors [114]. In addition, it should be noted that high-passage cell lines present limitations as models of tumors actually found in patients. Moreover, factors including molecular diversification of BDNF and controlling mechanisms related to trafficking and subcellular compartmentalization of different *Bdnf* mRNA forms may influence the response to BDNF/TrkB in different tumor models [115].

### 5.3. NT3, NT4/5, and TrkC

As with NGF/TrkA and BDNF/TrkB, immunohistochemical studies have found the expression of NT-3 and TrkC in subsets of MB samples. TrkC was observed in 17 of 20 MB tumors analyzed, and 3 of these tumors also co-expressed NT-3 [92], whereas other studies found TrkC in 48%, NT-3 in 9%, and NT-4/5 in 19% of 29 [93], TrkC in 71% of 22 [116], and TrkC in 52% of 21 [95] MB cases. Importantly, higher TrkC expression has been associated with a favorable outcome in MB. Thus, among 12 MB samples, all of which expressed mRNA encoding NT-3 and TrkC, patients with tumors expressing high levels of TrkC mRNA had significantly longer progression-free intervals and higher overall survival [117]. High TrkC mRNA expression was linked to a higher 5-year cumulative survival compared to patients with low expression (89% versus 46% respectively), with TrkC mRNA levels being the most powerful predictor of clinical outcome [118]. A more favorable outcome was also found among patients combining low MYC with high TrkC mRNA expression [119,120]. High TrkC expression is found particularly in MB tumors in the SHH subgroup [121]. Given that TrkC is expressed at higher levels in most mature cerebellar granule cells during CNS development, it has been proposed that MB tumors with a more favorable outcome could be derived from more differentiated cells [122]. TrkC expression has been incorporated as a predictor of MB patient survival in models combining clinical and biological markers [123], as well as in proposed risk stratification systems for MB [124].

When treated with NT-3, MB cells undergo apoptosis, and TrkC overexpression inhibits the growth of MB xenografts in nude mice. In addition, TrkC levels in individual cells from MB biopsies correlated with apoptosis [125], and TrkC mRNA levels were correlated to vincristine-induced apoptosis in DAOY and primary culture MB cells [126]. However, experimentally-induced TrkC overexpression in DAOY cells did not impact response to chemotherapy [121]. Tumors with reduced TrkC levels show decreased apoptosis in the Ptc+/- mouse model of MB [127]. NT-3-induced TrkC activation reduces MB cell invasion through a mechanism possibly involving heparanase inhibition [128]. Proteomic experiments have identified many proteins related to regulating gene expression, protein synthesis, apoptosis, proliferation, differentiation, migration, invasion, and cell metabolism as candidate effectors of NT-3 activation of TrkC in DAOY cells [129,130]. A truncated isoform of TrkC (t-TrkC), which is overexpressed and displays pro-proliferative actions in MB, and has its expression regulated by microRNAs miR-9 and miR-125a, has been identified [131].

### 5.4. p75NTR

In an immunohistochemical analysis of 167 MB samples, p75NTR was detected in 17% of classic MBs, in all of the desmoplastic (nodular) MBs, and 71% of those MBs with a significant desmoplastic component [132]. Immunostaining for p75NTR was also positive in nine (12%) of 75 MB tumors, being four classic, two desmoplastic, and three anaplastic MBs [133]. The ratio between TrkC and p75NTR expression correlates with meningeal spread in childhood MB samples [128], and p75NTR may play a role in a functional axis with heparanase in regulating MB invasion [134]. MB cells overexpressing p75NTR show increased apoptosis [135]. Blocking p75(NTR) proteolytic processing with a γ-secretase inhibitor impairs p75NTR-mediated migration, invasion, and spinal metastasis in experimental MB [136]. The presence of p75NTR in MB cell subpopulations may be related to a higher capacity for self-renewal [137], although other findings have suggested that p75NTR expression identifies lower self-renewing progenitors or stem cells and expression of SHH pathway genes in MB [138]. In fact, it was recently demonstrated, through immunohistochemical analysis and transcriptome data across 763 primary tumors, that p75NTR is a novel potential diagnostic and prognostic marker for SHH MB. The ERK/MAPK pathway was upregulated in p75NTR-positive tumors, and inhibiting MAPK signaling reduced stem/progenitor cell proliferation, survival, as well as migration [139]. A summary of findings from studies investigating neurotrophin signaling in MB is presented in Table 1.

### 5.5. Gene Expression Profile of Neurotrophins and Their Receptors in MB Primary Tumors

Analysis of data sets derived from 763 subgrouped primary MB samples from patients in previously published patient cohorts [11,15] and normal human cerebellum samples [140] revealed an increased expression of *NTRK1* in WNT tumors compared to all other MB groups (Figure 3), particularly in the WNT α subtype, common in young patients with monosomy of chromosome 6 and displaying good prognosis (Figure 4). High levels of *NTRK1* were also observed in the SHH β subtype (Figure 4A), which characterizes the poorest prognosis within the SHH subgroup. The impact of *NTRK1* expression on survival of patients with SHH MBs was evaluated using the Kaplan–Meier method, dividing the patients within groups displaying high and low expression of *NTRK1*. High expression of *NTRK1* was related to decreased overall survival (OS) probability in patients with SHH MB (Figure 5A).

High TrkC mRNA expression is frequent in SHH MB [121]. Our analysis showed *NTRK3* to be increased across all 4 subtypes of SHH tumors (Figure 4A). Evaluation of OS revealed that high *NTRK3* levels are associated with improved survival (Figure 3A) across all MB subgroups. In addition, patients with Group 4 MB displaying high expression of both *NTRK3* and NT-3 present a higher OS (Figure 5A). Consistent with these data, the hazard ratio analysis indicated that *NTRK3* can be considered a protective marker and associated with good prognosis.

Expression of BDNF was overall lower in MB in comparison with normal cerebellum (Figure 3C). Downregulation of BDNF may be a common feature not only in MB but among other brain tumor types [141]. WNT tumors showed the lowest expression of BDNF among MB subgroups (Figure 3C), whereas the highest expression of BDNF was observed in the SHHγ and SHHβ subtypes (Figure 4B). Both SHH subtypes are more prevalent in infants, and SHHβ tumors are frequently metastatic and have a worse overall survival compared with SHHγ [7,18]. Interestingly, SHHγ and SHHβ tumors display genes involved in developmental pathways, receptor tyrosine kinase signaling, bioelectrical activity and features of synaptic transmission [15].

Expression of BDNF across MB samples was investigated using the Kaplan–Meier method and showed that high expression was related to decreased OS (Figure 5A). Moreover, hazard ratio analysis suggested BDNF as a potential risk marker in MB (Figure 5B). We also detected decreased levels of *NTRK2* in subgroups SHH, Group 3, and Group 4 compared to normal cerebellum (Figure 3A), whereas increased levels were observed in Group 4 (Figure 3B), particularly in the Group 4α subtype (Figure 4A). Group 4 is the most prevalent subgroup comprising >40% of all MBs, and Group 4α are enriched for *MYCN* and *CDK6* amplifications. Genes linked to cell migration and neuronal development are also enriched in this subtype [18]. High levels of *NTRK1* and *NTRK2* correlated with decreased OS survival in MB-SHH patients (Figure 5A,B). A correlation map of NTRK and neurotrophin expression across MB samples is shown in Figure 5C, and Table 2 details statistical differences found among subgroups. Taken together, these data support the view that expression of neurotrophins and their receptors in MB has clinical implications, and these genes should be further investigated as potential biomarkers of molecular subgroups and subtypes of MB.

*NTRK1*, *NTRK2*, *NTRK3*, *NGF*, *BDNF*, and *NT-3* expression levels were examined in a previously described transcriptome data sets comprising a total of 986 patient samples and 9 normal cerebellum samples, from the Cavalli cohort [15] (*n =* 763 samples profiled on the Affymetrix Gene 1.1 ST array as previously described and normalized using the RMA method, and subgrouped using similarity network fusion, GSE85217), Pfister (*n =* 223 MB samples generated using Affymetrix Human Genome U133 Plus 2.0 Array), and Roth (*n =* 9 normal cerebellum samples, generated using Affymetrix Human Genome U133 Plus 2.0 Array, GSE3526). Expression of the 6 markers across all samples was normalized within the ’R2: Genomics Analysis and Visualization Platform (http://r2.amc.nl)’ and presented in box plot format as log2-transformed signal intensity. MB subtype classification was based on Cavalli et al. [28]. The number of patients with each MB subtype was as follows: 49 WNT α, 21 WNT β, 65 SHH α, 35 SHH β, 47 SHH γ, 76 SHH δ, 67 Group 3 α, 37 Group 3 β, 40 Group 3 γ, 98 Group 4 α, 109 Group 4 β, and 119 Group 4 γ. All subgroups and subtypes were compared using a Kruskal–Wallis test for significance and False Discovery Rate method. Statistical analyses were performed with the GraphPad prism 8.0 software; *p* ≤ 0.001 was considered statistically significant. Overall survival (OS) was measured from the time of initial diagnosis to the date of death or the date of last follow up, using combined OS and gene expression data from Cavalli et al. [15]. Expression of 6 markers across all samples was normalized within the ‘R2: Genomics Analysis and Visualization Platform (http://r2.amc.nl)’. Survival distribution was estimated according to the Kaplan–Meier method using a median cut-off and log-rank statistics; *p* ≤ 0.05 was considered statistically significant. Statistical analyses were performed with the GraphPad prism 8.0 software.

## 6. Conclusions

Oncogenic gene fusions involving the *NTRK* family have been recently identified across several tumor types and emerged as therapeutic targets. For example, among brain tumors, glioblastoma, pilocytic astrocytoma, and pontine glioma can show gene fusions involving *NTRK2* or *NTRK3* [63,87,142,143]. A number of small-molecule acting as pan-Trk inhibitors are currently being evaluated in clinical trials [144,145,146]. A recent phase 1 and 2 clinical study examining the effects of the pan-Trk inhibitor larotrectinib in children and adults with various types of peripheral solid cancers harboring *NTRK* gene fusions found pronounced and durable responses regardless of patient age or tumor type [147]. A case of a potent response to larotrectinib in a 3-year old female patient with a Trk fusion-driven pediatric high-grade glioma has been recently reported [148]. Although to date, oncogenic genetic alterations in the *NTRK* family have not been identified in MB, the findings reviewed here show accumulating evidence indicating that Trk expression can influence MB progression and should be further explored as a potential biomarker and therapeutic target. As reviewed above, the role of Trks in MB illustrates how childhood brain cancers can hijack molecular pathways involved in regulating neuronal survival, death, and differentiation during embryonic development. The differential roles of different subtypes of Trk receptors in MB, where stimulation of TrkA and TrkC can promote cell death whereas TrkB can likely display either pro- or antitumoral actions, makes it harder to predict what the clinical effects of pan-Trk inhibitors would be in MB patients. Further understanding of how neurotrophin signaling regulates MB tumor progression should increase our understanding of MB disease pathology and development of potential targeted therapeutic approaches.

The novel transcript analyses included in this review provide new insights into the role of neurotrophins and their receptors in MB. It is particularly worth highlighting that higher BDNF levels when all subgroups were analyzed together as well as higher *NTRK1* and *NTRK2* in SHH tumors were associated with reduced OS. These findings support the evidence from cultured cells and a mouse model indicating that TrkB inhibition can reduce MB cell viability and tumor growth.

## Figures and Tables

**Figure 1 cancers-12-02542-f001:**
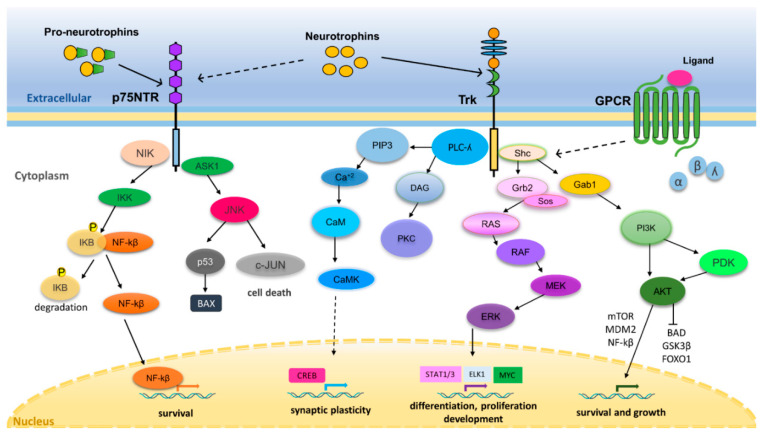
Signaling mechanisms mediating the actions of neurotrophins. Neurotrophin/Trk signaling involves neurotrophin binding to Trks, leading to receptor dimerization, autophosphorylation of tyrosine residues, and the recruitment of cytosolic adaptor proteins such as Src homology collagen protein (Shc). Shc recruits the adaptor growth factor receptor-bound protein 2 (GRB2) which is linked to the Ras exchange factor son of sevenless (SOS) leading to the activation of a RAS- MAPK pathway (RAS-RAF-MEK-ERK). Activated extracellular-regulated kinase (ERK) translocates to the nucleus and transactivates transcription factors such as STAT1/3, Elk1, and Myc, modulating gene expression to induce proliferation, differentiation or development. Shc can also recruit GRB2-associated-binding protein 1 (Gab1), driving activation of the PI3K-PDK1-Akt cascade. Phosphorylated Akt can regulate cell survival, growth, and angiogenesis via activation or inactivation of downstream targets. Akt can play an anti-apoptotic role through inactivation of Forkhead box protein O1 (FOXO1) transcription factor, Bcl-2-associated death promoter (BAD) and glycogen synthesis kinase (GSK-3β). Akt activates several proteins important for cell survival such as mammalian target of rapamycin (mTOR), murine double minute 2 (MDM2) and NF-k β. Phosphorylation of PLCγ by Trk receptors enables catalysis of phosphatidylinositol 4,5-biphosphate (PIP 2) cleavage to diacylglycerol (DAG) and inositol triphosphate (IP 3), which releases calcium from intracellular stores, activating calmodulin (CaM) and Ca^2+^ /CaM-dependent protein kinases (CaMKs). Together, these signaling molecules stimulate multiple intracellular enzymes that regulate the activity of transcription factors, such as cAMP response element-binding protein (CREB) and ion channels involved in the control of synaptic plasticity. Trk transactivation may be mediated by G protein-couple receptors (GPCRs). GPCR-activated members of c-Src family induce a neurotrophin-independent transactivation of a Trk via trans-phosphorylation of cytosolic tyrosines, which provide docking sites for triggering intracellular signaling cascades like PI3K-PDK1-Akt and MAPK. Binding of neurotrophins to p75NTR can activate either the NF-kB signaling pathway producing prosurvival signals, or c-Jun N-terminal kinase (JNK) signaling to induce cell death. P75NTR can activate the kinase NIK, which in turn stimulates IKK complexes that phosphorylate IKB, leading to its ubiquitination and proteasomal processing and subsequent releasing of NF-kB. This creates transcriptionally competent NF-κB complexes that translocate to the nucleus and induce the expression of survival genes. Neurotrophin binding to p75NTR can also induce activation of apoptosis signal-regulating kinase 1 (ASK1), which, in turn, activates JNK. JNK itself, or via c-JUN phosphorylation, stimulates p53-mediated apoptosis by regulating its targets such as Bcl-2-associated X protein (BAX). Pro-neurotrophins can also bind to p75NTR to initiate cell apoptosis via the JNK signaling pathway [31,33,34,42].

**Figure 2 cancers-12-02542-f002:**
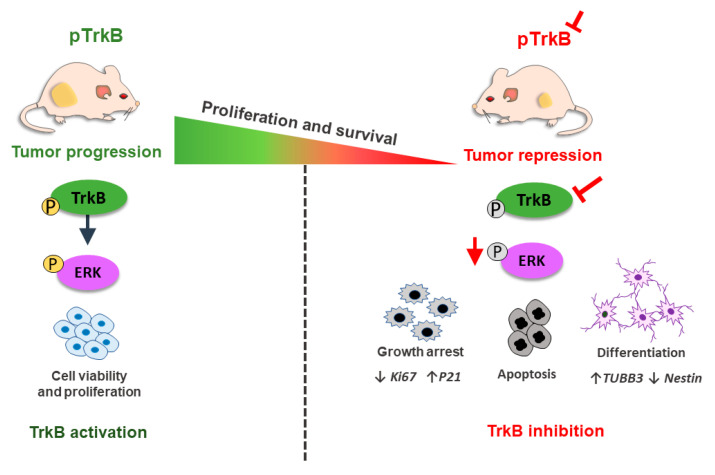
TrkB inhibition decreases proliferation and survival pathways leading to a reduction of tumor growth and increased apoptosis and differentiation features in medulloblastoma (MB) cells. Scheme depicting mouse xenografted with MB cells. In the absence of TrkB antagonist, MB cells continue to grow at high proliferation rates. Treating mice with TrkB antagonist, ANA-12, promotes delay in tumor growth in vivo and cellular changes consistent with growth arrest, apoptosis and differentiation mediated by downregulation of ERK pathway, decreased KI67 and Nestin expression markers and increased expression of p21 and TUBB3 genes (modified from [111]).

**Figure 3 cancers-12-02542-f003:**
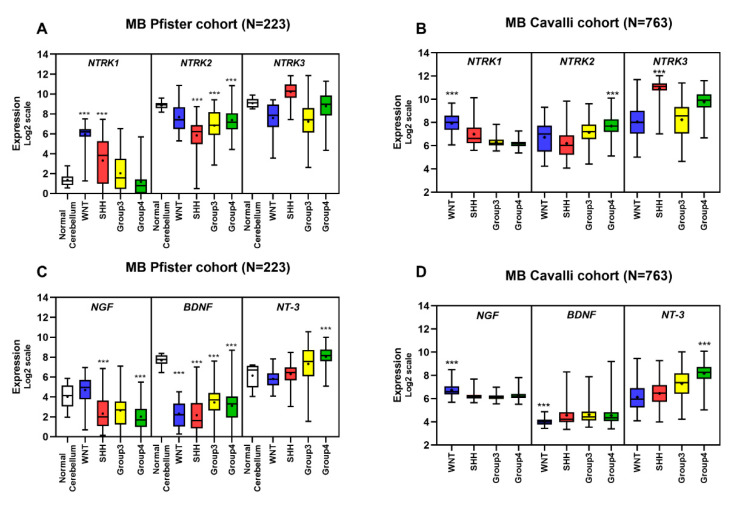
Transcript levels of neurotrophins and their receptors in tumors across the 4 MB molecular variants compared to expression in normal human cerebellum. Expression levels were examined in previously described transcriptome data sets comprising samples from Pfister (*n =* 223 MB samples; [22]), Roth (*n =* 9 normal cerebellum samples; [11]), and Cavalli et al. [15] (*n =* 763 MB samples). Expression of NTRKs (**A**,**B**) and neurotrophins (**C**,**D**) across all samples is presented in boxplot format as log2-transformed signal intensity. All subgroups were compared using a Kruskal–Wallis test followed by the False Discovery Rate method. Data are shown as median and whiskers: min to max. Statistical differences in comparison to normal cerebellum are shown in panels A and C, and differences between all subgroups in panels B and D; *** *p* ≤ 0.001 for significance.

**Figure 4 cancers-12-02542-f004:**
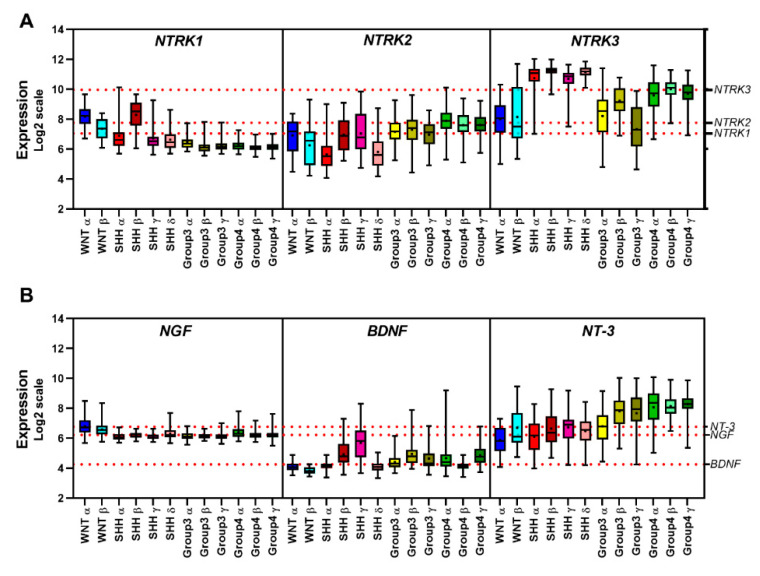
Transcript levels of neurotrophins and their receptors in tumors across the 12 MB molecular subtypes. Expression levels were examined in previously described transcriptome data set comprising tumor samples from patients in the Cavalli cohort (*n =* 763 MB samples) [15]. Expression of NTRKs (**A**) and neurotrophins (**B**) across all samples is presented in boxplot format as log2-transformed signal intensity. Comparisons among subgroups were performed using a Kruskal–Wallis test followed by the False Discovery Rate method. Data are shown as median and whiskers: min to max; the red dotted line displays the median of expression of each gene according to MB subtype. Sample characteristics and general methods for expression profiling analyses were as described in the legend for Figure 3.

**Figure 5 cancers-12-02542-f005:**
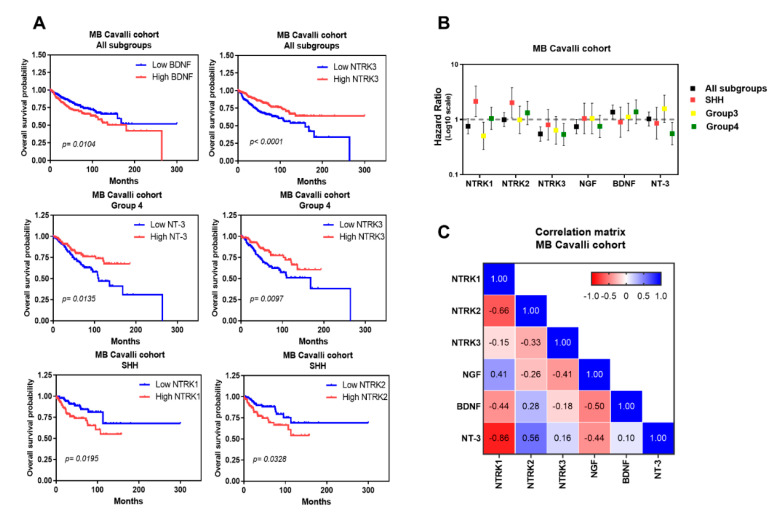
Prognostic value of neurotrophins and their receptors in MB patients. Kaplan–Meier overall survival curve from MB patients according to the expression level neurotrophins and their receptors (**A**). Kaplan–Meier plot of MB overall survival stratified by the median of expression for each marker in the Cavalli cohort [15] are classified into low or high expression levels. The statistical significance was determined using a log-rank test (*p* ≤ 0.05). Log of Hazard Ratios of NTRKs and neurotrophins (**B**). Markers with a hazard ratio smaller than 1.00 represent ‘‘protective markers’’ and those with hazard ratios larger than 1.00 represent ‘‘risk markers’’. (**C**). Heatmap of the correlation matrix of the NTRKs and neurotrophins in all molecular subgroups from the cohort. The correlation coefficient is colour-coded from red (−1) representing a negative correlation to blue (1) representing a positive correlation. Sample characteristics and general methods for expression profiling analyses were as described in the legend for Figure 3.

**Table 1 cancers-12-02542-t001:** Summary of studies examining the involvement of neurotrophin receptors in MB.

Receptor	TrkA	TrkB		TrkC	Truncated TrkC	p75NTR	
Type of alteration	Expression and activation by NGF [100]	Expression and activation by BDNF [93]	Inhibition [110,111]	Overexpression and NT-3 activation [117]	Overexpression [131]	Expression [137]	Inhibition [136]
Main functional effect	↑Cell death (apoptosis [101] or micropinocytosis [106]) ↓Proliferation and ↑differentiation [94]	↓Cell viability [109] BDNF+ HDACi ↓ Cell viability [112]	↓Cell viability, proliferation and survival [110] ↑Apoptosis and differentiation↓ Subcutaneous tumor growth in nude mice [111]	↑Apoptosis and differentiation [125]	↑Proliferation Targeted by miR-9 and miR-125a to inhibit cell proliferation [131]	Marker for SHH progenitor/stem cells [137]	↓Migration, proliferation and spinal metastasis [136]
Clinical evidence	Apoptotic index and neuronal differentiation [95]	Unknown	Unknown	Higher overall survival [118] Favorable outcome [120] High expression in SHH MB [121]	Unknown	Potential diagnostic and prognostic marker for SHH group [138,139]	Unknown

↑ increase; ↓ decrease.

**Table 2 cancers-12-02542-t002:** Differences between high and low gene expression levels of neurotrophins and their receptors and their respective Hazard Ratio values across the 4 MB molecular variants in tumors from patients from the Cavalli cohort [15].

Marker	Subgroups	*p*-Value (High vs Low Gene Expression Levels)	Hazard Ratio	95% CI
***NTRK1***	All	0.0727	0.7512	0.55 to 1.02
	WNT	0.2113	0.1654	0.009 to 2.77
	**SHH**	**0.0195**	**2.142**	**1.13 to 4.05**
	Group3	0.0728	0.5038	0.28 to 0.89
	Group4	0.8478	1.046	0.65 to 1.66
***NTRK2***	All	0.7805	1.045	0.76 to 1.42
	WNT	0.0531	0.1022	0.01 to 1.03
	**SHH**	**0.0328**	**2.014**	**1.06 to 3.81**
	Group3	0.7083	0.987	0.55 to 1.75
	Group4	0.254	1.321	0.80 to 2.16
***NTRK3***	**All**	**<0.0001**	**0.5242**	**0.38 to 0.71**
	WNT	0.0574	0.1067	0.01 to 1.07
	SHH	0.4931	0.8012	0.42 to 1.51
	Group3	0.1206	0.6388	0.35 to 1.13
	**Group4**	**0.0097**	**0.5352**	**0.33 to 0.85**
***NGF***	All	0.0784	0.743	0.55 to 0.99
	WNT	0.9105	0.8902	0.08 to 9.41
	SHH	0.8937	1.044	0.55 to 1.97
	Group3	0.5987	0.8576	0.48 to 1.52
	Group4	0.2205	0.7505	0.47 to 1.19
***BDNF***	**All**	**0.0104**	**1.503**	**1.10 to 2.05**
	WNT	0.1552	0.08509	0.008 to 0.86
	SHH	0.7331	0.8956	0.47 to 1.69
	Group3	0.7175	1.111	0.62 to 1.97
	Group4	0.1912	1.377	0.83 to 2.26
***NT-3***	All	0.5943	1.088	0.79 to 1.48
	WNT	0.9406	0.9141	0.08 to 9.74
	SHH	0.6472	0.8559	0.44 to 1.66
	Group3	0.1234	1.571	0.88 to 2.78
	**Group4**	**0.0135**	**0.5557**	**0.34 to 0.88**

Bold face representing variables with *p*-values ≤ 0.05. CI, confidence interval.

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
