# Peer review of "Neurotrophin Signaling in Medulloblastoma"

_cancers, 2020, doi:10.3390/cancers12092542_

Round 1
Reviewer 1 Report
A review on the role of “Neurotrophin Signaling in Medulloblastoma” should be of interest to the readership ofCancers.
Below are a few comments on the review the authors may consider:
-Though grammatically correct, the structure of many sentences in the abstract makes them hard to read. Authors may consider simplifying them and focus on the main observations and take-home messages of the review. For example, the various names of the genes/proteins ligands/receptors and their pairing do not need to be discussed in the abstract.
-“ …that higher levels of NTRK2 are associated with reduced overall survival (OS) of patients with a specificsubgroup of MB tumors”.
Which subgroups?
-“Nestin-positive progenitors (NEPs)“
Consider Nestin-expressing progenitors (NEPs)
-Authors should comment on the limitations of the studies performed with high-passage cell lines, which are not very representative of the primary MB tumors.
-Authors should comment on the mutational status (deletion, fusion, etc.) of Neurotrophin Signaling genes in MB. If none are observed, the authors should state so. I now see this is mentioned in the conclusion. A bit too late in the narrative. Consider stating it earlier.
-in Table 1, “expression and activation of NGF” does not need to be bold, does it?
-“Analysis of data sets derived from 763 subgrouped primary MB samples from patients in previously 2 published patient cohorts [11, 15] and normal human cerebellum samples [139] revealed an increasedexpression of NGF and NTRK1 in WNT tumors compared to all other MB groups (Figure 3), particularly in 4 theWNT α subtype, common in young patients with monosomy of chromosome 6 and displaying good 5 prognosis(Figure 4).”
I agree for NTRK1. However, in my opinion, visual inspection of the figures/data does not show a strongdifferential expression for NGF in any specific subtype in comparison to other subtypes or normals (not observed in Pfister and not a large difference in spite of P value in Cavalli). Any comment on NGF expression and its effect on OS?
-“ High TrkC mRNA expression appears to be frequent in SHH MB” Many statements are hypothetical as oppose to affirmative. Verbs such as “appear, seem, may” are used often.
-“ Further understanding of how neurotrophin signaling regulates MB tumor progression should increase our understanding of MB disease pathology and development of potential targeted therapeutic approaches.” Authors claim that more studies should be done but they do not make any suggestions. Any comments on theavailability of KO mice (or lack of) for these Neurotrophin Signaling genes and the opportunity to study their KOin genetically engineered mouse models of MB?
Author Response
Response to Reviewer 1
Manuscript ID: cancers-912332
Entitled: Neurotrophin Signaling in Medulloblastoma
First author: Amanda Thomaz
Corresponding author: Rafael Roesler
REVIEWER #1
- Reviewer’s comment: Though grammatically correct, the structure of many sentences in the abstract makes them hard to read. Authors may consider simplifying them and focus on the main observations and take-home messages of the review. For example, the various names of the genes/proteins ligands/receptors and their pairing do not need to be discussed in the abstract.
Response: The Abstract has been edited to address the Reviewer’s comment (lines 28 and 29).
- Reviewer’s comment: -“ …that higher levels of NTRK2 are associated with reduced overall survival (OS) of patients with a specific subgroup of MB tumors”.
Which subgroups?
Response: The Abstract has been modified in order to address the Reviewer’s comment. It now reads as follows (lines 37 and 38): “that higher levels of NTRK1 or NTRK2 are associated with reduced overall survival (OS) of patients with SHH MB tumors.”
- Reviewer’s comment: -“Nestin-positive progenitors (NEPs)“
Consider Nestin-expressing progenitors (NEPs)
Response: The change was made, according to the Reviewer’s suggestion (line 83).
- Reviewer’s comment: Authors should comment on the limitations of the studies performed with high-passage cell lines, which are not very representative of the primary MB tumors.
Response: A statement was included to address the Reviewer’s comment, as follows: “In addition, it should be noted that high-passage cell lines present limitations as models of tumors actually found in patients.” (lines 279-280).
- Reviewer’s comment: Authors should comment on the mutational status (deletion, fusion, etc.) of Neurotrophin Signaling genes in MB. If none are observed, the authors should state so. I now see this is mentioned in the conclusion. A bit too late in the narrative. Consider stating it earlier.
Response: We have included a statement earlier in the paper (section 4) to address the Reviewer’s comment, as follows (lines 218-219): “To date, such genetic alterations in NTRK genes have not been reported in MB.”
- Reviewer’s comment: -in Table 1, “expression and activation of NGF” does not need to be bold, does it?
Response: The bold mark has been removed according to the Reviewer’s observation (see Table 1).
- Reviewer’s comment: Analysis of data sets derived from 763 subgrouped primary MB samples from patients in previously 2 published patient cohorts [11, 15] and normal human cerebellum samples [139] revealed an increased expression of NGF and NTRK1 in WNT tumors compared to all other MB groups (Figure 3), particularly in 4 theWNT α subtype, common in young patients with monosomy of chromosome 6 and displaying good 5 prognosis(Figure 4).”
I agree for NTRK1. However, in my opinion, visual inspection of the figures/data does not show a strong differential expression for NGF in any specific subtype in comparison to other subtypes or normals (not observed in Pfister and not a large difference in spite of P value in Cavalli). Any comment on NGF expression and its effect on OS?
Response: The mention to NGF pointed out by the Reviewer was removed (line). NGF expression in relation to OS is not discussed given that no significant effect was found.
- Reviewer’s comment: “High TrkC mRNA expression appears to be frequent in SHH MB” Many statements are hypothetical as oppose to affirmative. Verbs such as “appear, seem, may” are used often.
Response: The whole text has been revised to change hypothetical statements to more affirmative ones, as suggested by the Reviewer.
- Reviewer’s comment: “Further understanding of how neurotrophin signaling regulates MB tumor progression should increase our understanding of MB disease pathology and development of potential targeted therapeutic approaches.” Authors claim that more studies should be done but they do not make any suggestions. Any comments on theavailability of KO mice (or lack of) for these Neurotrophin Signaling genes and the opportunity to study their KOin genetically engineered mouse models of MB?
Response: The use of knockout mouse models in cancer studies is often complicated by the difficulty of growing human cancer cells as xenografts in non-immunosuppressed mice. There are genetic mouse models of MB, however these models have not been associated to genetic manipulation of components of neurotrophin signaling.

Reviewer 2 Report
Thomaz et al. reviewed the neurotrophin signaling in Medulloblastoma (MB). It’s already well-categorized and adequately summarized. This review paper also contains timely and interesting perspectives; thus, I hope it to be published in this journal.
Author Response
Response to Reviewer 2
Manuscript ID: cancers-912332
Entitled: Neurotrophin Signaling in Medulloblastoma
First author: Amanda Thomaz
Corresponding author: Rafael Roesler
REVIEWER #2
- Reviewer’s comment: Thomaz et al. reviewed the neurotrophin signaling in Medulloblastoma (MB). It’s already well-categorized and adequately summarized. This review paper also contains timely and interesting perspectives; thus, I hope it to be published in this journal.
Response: We thank the Reviewer and are glad that the Reviewer’s opinion is that our paper should be accepted as it states.

Reviewer 3 Report
The manuscript by Thomaz et al. reviews the role of neurotrophins and their receptors signaling in medulloblastoma. Neurotrophins and their receptors have been long recognized to play an important role in the development and maintenance of the nervous system, but the first Trk receptor was identified originally as a rearrangement in a colon carcinoma. In addition, it has been described the role of neurotrophin receptor rearrangements in other cancer types. Now Thomaz et al. reviews the bibliography related to neurotrophins and their receptors in medulloblastoma, which affects mainly to children. The authors stressed the correlation between TrkB/BDNF axis and reduced overall survival in medulloblastoma patients. The manuscript is well-written and the authors performed a thorough analysis of the literature.
Main concerns:
- Line 167: The reference “Pulciani et al 1982” is wrong regarding the discovery of the NTRK1 gene fused to non-muscle tropomyosin. The correct reference is Martin-Zanca et al. 1986.
- Line 297. “of NT-3 and TrkB in subsets….” Is TrkB correct? Should not be TrkC?
Author Response
Response to Reviewer 3
Manuscript ID: cancers-912332
Entitled: Neurotrophin Signaling in Medulloblastoma
First author: Amanda Thomaz
Corresponding author: Rafael Roesler
REVIEWER #3
- Reviewer’s comment: The manuscript by Thomaz et al. reviews the role of neurotrophins and their receptors signaling in medulloblastoma. Neurotrophins and their receptors have been long recognized to play an important role in the development and maintenance of the nervous system, but the first Trk receptor was identified originally as a rearrangement in a colon carcinoma. In addition, it has been described the role of neurotrophin receptor rearrangements in other cancer types. Now Thomaz et al. reviews the bibliography related to neurotrophins and their receptors in medulloblastoma, which affects mainly to children. The authors stressed the correlation between TrkB/BDNF axis and reduced overall survival in medulloblastoma patients. The manuscript is well-written and the authors performed a thorough analysis of the literature.
Main concerns:
Line 167: The reference “Pulciani et al 1982” is wrong regarding the discovery of the NTRK1 gene fused to non-muscle tropomyosin. The correct reference is Martin-Zanca et al. 1986.
Response: That reference has been replaced by the correct one in the revised manuscript, as suggested by the Reviewer (reference 42).
- Reviewer’s comment: Line 297. “of NT-3 and TrkB in subsets….” Is TrkB correct? Should not be TrkC?
Response: Yes, it was been corrected to TrkC in the revised manuscript.
